# Long-term Cross Adversarial Training: A Robust Meta-learning Method for Few-shot Classification Tasks

**Fan Liu** [1]  **Shuyu Zhao** [1]  **Xuelong Dai** [1]  **Bin Xiao** [1]

## Abstract

Meta-learning model can quickly adapt to new tasks using few-shot labeled data. However, despite achieving good generalization on few-shot classification tasks, it is still challenging to improve the adversarial robustness of the meta-learning model in few-shot learning. Although adversarial training (AT) methods such as Adversarial Query (AQ) can improve the adversarially robust performance of meta-learning models, AT is still computationally expensive training. On the other hand, meta-learning models trained with AT will drop significant accuracy on the original clean images. This paper proposed a meta-learning method on the adversarially robust neural network called Long-term Cross Adversarial Training (LCAT). LCAT will update meta-learning model parameters cross along the natural and adversarial sample distribution direction with long-term to improve both adversarial and clean few-shot classification accuracy. Due to cross-adversarial training, LCAT only needs half of the adversarial training epoch than AQ, resulting in a low adversarial training computation. Experiment results[2] show that LCAT achieves superior performance both on the clean and adversarial few-shot classification accuracy than SOTA adversarial training methods for meta-learning models.

## 1. Introduction

Meta-learning only needs few-shot data that can quickly adapt to even unseen tasks and has been widely applied to many fields (Haoxiang Wang, 2021). However, recent research shows that meta-learning based-models are also vulnerable to adversarial examples (Goldblum et al., 2020; Croce & Hein, 2019). Adversarial training trained on the large-scale dataset can improve the adversarial robustness of deep neural networks (Rice et al., 2020). However, how to train on few-shot tasks to robustly defend adversarial examples, fast adapt to other tasks is still challenging.

Recent robust meta-learning work called Adversarial Query (AQ) (Goldblum et al., 2020) demonstrates that adversarial training only during the query step can compromise the superior adversarially robust performance compared to traditional adversarial training (Madry et al., 2017b). However, AQ is still computationally expensive training throughout the whole query step. In addition, AQ will drop significant accuracy on the original clean image distribution, which means meta-learning models cannot adapt well to other unseen tasks.

This paper proposes a new robust meta-learning method called Long-term Cross Adversarial Training (LCAT). In detail, LCAT will make the meta-learning model's parameter $\theta$ cross along the natural and adversarial sample distribution direction. LCAT is a model-agonist meta-learning method, can adversarially adapt to few-shot classification tasks, i.e., perform well on the original clean sample distribution and robust to adversarial examples. Our solved problem can be formulated as Eq. (1), where $T(x, y)$ is training data distribution. $\theta$ is meta-learning model $f_\theta$'s parameters. $A$ is the fine-tuning algorithm during the inner loop for meta-learning. $\epsilon$ is the maximum bound for adversarial perturbation. $\mathbb{L}$ is the loss function.

$$\min_{\theta} \mathbb{E}_{T(x,y)} \left[ \max_{\|\epsilon\|_p \leq \delta} \mathbb{L}(f_{A(\theta, T)}(x + \epsilon), y) \right] \quad (1)$$

LCAT only needs half of the adversarial training epoch via cross adversarial training and significantly improves the robust performance compared to the the-state-of-art adversarial training AQ for meta-learning models. The proposed method LCAT is illustrated in Fig. 1. Our contribution can be summarized as follows:

- We propose an adversarially robust meta-learning

---

[1]Department of Computing, The Hong Kong Polytechnic University, Hong Kong. Correspondence to: Fan Liu <cs-fan.liu@connect.polyu.hk>.

*Accepted by the ICML 2021 workshop on A Blessing in Disguise: The Prospects and Perils of Adversarial Machine Learning.* Copyright 2021 by the author(s).

[2]The code to reproduce all the experimental results is available in https://github.com/Gnomeek/Long-term-Cross-Adversarial-Training.

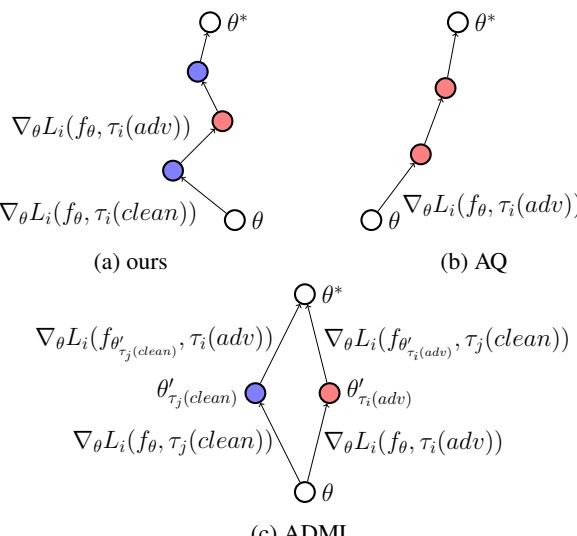

(a) ours        (b) AQ

(c) ADML

*Figure 1.* An illustration of our proposed method. LCAT will make the meta-learning model's parameter $\theta$ cross along the natural and adversarial sample distribution direction. A precious state-of-the-art adversarial training method AQ only makes models' parameters updated on the adversarial sample space. Another adversarial training method named ADML (Yin et al., 2018) is only designed for MAML (Finn et al., 2017) and cannot well against stronger attackers such as PGD (Madry et al., 2017a). Compared to AQ, our LCAT only needs half of the adversarial training epoch has higher adversarial and clean few-shot classification accuracy.

method called LCAT. LCAT will update meta-learning model parameters cross along the natural and adversarial sample distribution direction with long-term to improve both adversarial and clean few-shot classification accuracy.

- LCAT is a model-agonist meta-learning method, which can improve the adversarial robustness of meta-learning models. In addition, LCAT only needs half of the adversarial training epoch compared to AQ via cross adversarial training, resulting in a low adversarial training computation.

- Experimental results show that LCAT improves both clean and adversarial accuracy compared to SOTA method AQ. i.e. improve 9.7% clean and 2.88% adversarial few-shot classification accuracy compared to previous best results in MetaOptNet (Lee et al., 2019) corresponding to 5-way 5-shot on MiniImageNet dataset.

## 2. Related Work

There're several methods to train a robust Deep Learning model in normal scenarios rather than a meta-learning model, including defensive distillation (Papernot et al.,

2016), adversarial training (Madry et al., 2017a; Su et al., 2016), feature denoising (Xie et al., 2019), TRADES theoretically principled the trade-off between accuracy and robustness (Zhang et al., 2019), and adversarial example detection (Hendrycks & Gimpel, 2016; Xu et al., 2017; Gong et al., 2017; Grosse et al., 2017). There are many adversarial robust deep neural network applications, including (Pang et al., 2020; Dong et al., 2020; Feng et al., 2021; Hendrycks et al., 2020; Dai et al., 2018; Pang et al., 2021; Dong et al., 2021).

Some works published in recent years discussed the robustness of the meta-learning model from different aspects. Yin et al. (2018) introduce a robust variant of MAML (Finn et al., 2017) called ADML, which is the first attempt to achieve robustness on the meta-learning model. However, ADML needs lots of computational power while the attacker used in their experiments is relatively weak. Goldblum et al. (2020) present a method to integrate MAML with adversarial training, called adversarial querying(AQ), which is the state-of-the-art approach in this domain. We choose AQ as one of the baselines in our experiments. In (Wang et al., 2021), they build a principled robustness-regularized meta-learning framework, which can be treated as a generalized AQ model.

## 3. Proposed Method: LCAT

Our goal is to make meta-learning models perform well both on clean and adversarial few-shot classification tasks. We now formulated the problem we want to solve, which can be seen in Eq. (1).

**Cross adversarial training** The proposed method called Long-term cross adversarial training (LCAT) is illustrated in Fig. 1. The LCAT across along the direction of the clean and adversarial direction. In the clean meta-update step, the LCAT will optimize the meta-learning model $f_\theta$ in the clean batch of tasks. This step aims to give the meta-model a good initial start which quickly adapts to clean few-shot classification tasks. In the adversarial meta-update step, the LCAT will optimize $f_\theta$ in the adversarial batch of tasks, which can help the meta-learning model against adversarial attacks.

It is important to notice that LCAT is model-agnostic. Our adversarial training method can help the meta-learning model improve significance accuracy on the clean image distribution and help the meta-learning model improve adversarially robust accuracy with a low computation compared to a previous SOTA adversarial training AQ.

The method of LCAT training method is as follows.

**STEP 1** Generate a batch of tasks according to the distribution of tasks $\rho(\tau)$ (Goldblum et al., 2020).

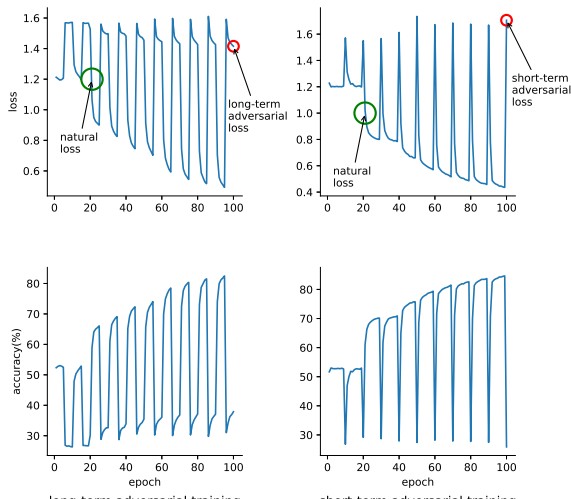

*Figure 2.* The comparison between long-term and short-term adversarial training.

**STEP 2** Fine-tune each task $\{\tau_i\}_{0 \leq i < n}$ based on the fine-tuning algorithm $A(\theta, T)$ .

**STEP 3** Make model parameter theta update along with the clean sample space with long-term $T$ based on Eq. (2).

$$\theta \leftarrow \theta - \frac{\lambda}{n} \sum_{0 \leq j < T} \sum_{\tau_i \sim \rho(\tau)} \nabla_\theta \mathbb{L}(f_\theta^{(j)}, \tau_i) \qquad (2)$$

Where $\theta$ is the meta-leaning model parameter, and $f_\theta^{(j)}$ is the meta-learning model at $j$th term. $\lambda$ is the learning rate.

**STEP 4** Compute adversarial examples of tasks $\{\tau_i'\}$ based on attacker (Madry et al., 2017a) within a $\epsilon$ ball and then make the model update in the space of adversarial examples for long-term T based on Eq. (3).

$$\theta \leftarrow \theta - \frac{\lambda}{n} \sum_{T \leq j' < 2T} \sum_{\tau_i' \sim \rho(\tau')} \nabla_\theta \mathbb{L}(f_\theta^{(j')}, \tau_i') \qquad (3)$$

The detail of the LCAT training method is shown in Alg. 1.

**Why need long-term cross training?** Short-term cross adversarial training cannot give suitable initial parameters for meta-learning models, as shown in Fig. 2. Long-term cross adversarial training can make the model's parameters go deeper in the adversarial sample space, resulting in good adversarial training. The model can not be well trained in the adversarial sample space corresponding to short-term adversarial training. In Tab. 2, we test the short-term adversarial training. In detail, we set ten epochs as a loop, the first nine epochs are natural training, and the last epoch is adversarial training. Fig. 2 and Tab. 2 show the training results. The test results show that the model trained with

short-term has low adversarial accuracy compared to long-term, cannot sufficiently defend adversarial samples. In addition, to avoid long-term cross-training appearing in the clean and adversarial sample space to go deeper and lead to poor performance of the model in few-shot classification, we also added the denoise module (Xie et al., 2019) for the meta-learning model. The Sec. 5 shows the effectiveness of the denoise module.

---

**Algorithm 1** LCAT training method

**Input:**
    Meta-learning model, $f_\theta$, learning rate, $\lambda$, fine-tune algorithm A, tasks distribution $\rho(\tau)$, long-term $T$;
1:  Initialize all parameters;
2:  **while** not done **do**
3:      Sample batch of tasks, $\{\tau_i\}_{0 \leq i < n}$;
4:      **for** $j = 0$; $j < T$; $j + +$ **do**
5:          **for** $i = 0$; $i < n$; $i + +$ **do**
6:              $\theta = A(\theta, \tau_i)$, fine tune on task $\tau_i$;
7:              Compute updated gradient $\nabla_\theta \mathbb{L}(f_\theta^{(j)}, \tau_i)$;
8:          **end for**
9:      **end for**
10:     $\theta \leftarrow \theta - \frac{\lambda}{n} \sum_{0 \leq j < T} \sum_{\tau_i \sim \rho(\tau)} \nabla_\theta \mathbb{L}(f_\theta^{(j)}, \tau_i)$;
11:     **for** $j' = T$; $j' < 2T$; $j' + +$ **do**
12:         **for** $i = 0$; $i < n$; $i + +$ **do**
13:             $\theta = A(\theta, \tau_i)$, fine tine on task $\tau_i$;
14:             Generate adversarial batch of tasks, $\{\tau_i'\}_{0 \leq i < n}$ $\nabla_\theta \mathbb{L}(f_\theta, \tau_i')$;
15:             Compute updated gradient $\nabla_\theta \mathbb{L}(f_\theta^{(j')}, \tau_i')$;
16:         **end for**
17:     **end for**
18:     $\theta \leftarrow \theta - \frac{\lambda}{n} \sum_{T \leq j' < 2T} \sum_{\tau_i' \sim \rho(\tau')} \nabla_\theta \mathbb{L}(f_\theta^{(j')}, \tau_i')$;
19: **end while**

---

## 4. Experiments

### 4.1. Setup

**Baselines** To test the effectiveness of LCAT, we compared our adversarial training with 1) Adversarial Training (AT) (Madry et al., 2017b), a directly robust adversarial training method applied in the meta-learning method. 2) The Adversarial Query (AQ) (Goldblum et al., 2020) is a SOTA adversarial training method that only needs adversarial training in the query step. 3) Adversarial Query plus TRADES (Zhang et al., 2019) loss function can be a trade between natural accuracy and robust accuracy. We call this method AQ+TRADES which is suitable as a SOTA method compared to LCAT.

The learning rate is set to 0.1. The epoch and the batch size are set to 50 and 8. We use Adam (Kingma & Ba, 2014) to optimize the meta-learning models, which has been proved to obtain better performance than SGD (see Appendix A.3).

*Table 1.* Experiments results of 50 epoch on MiniImageNet dataset with three models, $Acc_{nat}$ and $Acc_{adv}$ stand for natural and robust accuracy of the model, respectively. Robust accuracy is computed for a 20-step PGD attack on the model. Top accuracy scores and the shortest training time are shown in bold font.

| DATASET | METHODS | PROTONET | | | R2D2 | | | METAOPTNET | | |
|---|---|---|---|---|---|---|---|---|---|---|
| | | $Acc_{nat}$ | $Acc_{adv}$ | TIME | $Acc_{nat}$ | $Acc_{adv}$ | TIME | $Acc_{nat}$ | $Acc_{adv}$ | TIME |
| MINIIMAGENET | AT (5WAY-1SHOT) | 30.19 % (0.43 %) | 19.15 % (0.36 %) | 17.7H | 26.84 % (0.40 %) | 18.86 % (0.36 %) | 19.7H | 33.27 % (0.50 %) | 13.25 % (0.36 %) | 24.8H |
| | AQ (5WAY-1SHOT) | 29.59 % (0.43 %) | 19.78 % (0.37 %) | 8.4 H | **27.05 % (0.39 %)** | 19.06 % (0.35 %) | 10.1H | 32.43 % (0.49 %) | 13.11 % (0.35 %) | **9.4H** |
| | AQ+TRADES (5WAY-1SHOT) | 29.61 % (0.43 %) | 20.48 % (0.37 %) | 10.2H | 26.85 % (0.39 %) | **19.34 % (0.35 %)** | 12.8H | 29.23 % (0.44 %) | 16.75 % (0.36 %) | 12.1H |
| | LCAT (OURS, 5WAY-1SHOT) | **32.55 % (0.49 %)** | 19.72 % (0.41 %) | **6.6H** | 26.74 % (0.40 %) | 18.73 % (0.35 %) | **8.4H** | **34.88 % (0.52 %)** | 14.25 % (0.37 %) | 9.6H |
| | LCAT+TRADES (OURS, 5WAY-1SHOT) | 30.71 % (0.46 %) | **20.56 % (0.39 %)** | 7.7H | 26.25 % (0.39 %) | 19.29 % (0.33 %) | 10.3H | 30.15 % (0.46 %) | **18.35 % (0.39 %)** | 10.3H |
| | AT (5WAY-5SHOT) | 41.09 % (0.47 %) | 25.85 % (0.41 %) | 17.7H | 34.95 % (0.42 %) | 23.48 % (0.39 %) | 19.7H | 45.82 % (0.50 %) | 18.86 % (0.44 %) | 24.8H |
| | AQ (5WAY-5SHOT) | 39.89 % (0.46 %) | 26.01 % (0.41 %) | 8.4H | 35.06 % (0.42 %) | 23.89 % (0.39 %) | 10.1H | 45.44 % (0.51 %) | 19.17 % (0.44 %) | **9.4H** |
| | AQ+TRADES (5WAY-5SHOT) | 39.05 % (0.46 %) | 26.71 % (0.42 %) | 10.2H | 33.23 % (0.40 %) | 23.77 % (0.38 %) | 12.8H | 38.13 % (0.48 %) | 21.87 % (0.41 %) | 12.1H |
| | LCAT (OURS, 5WAY-5SHOT) | **44.81 % (0.52 %)** | 27.89 % (0.47 %) | **6.6H** | **39.18 % (0.49 %)** | 24.84 % (0.45 %) | **8.4H** | **47.83 % (0.53 %)** | 21.34 % (0.45 %) | 9.6H |
| | LCAT+TRADES (OURS, 5WAY-5SHOT) | 42.14 % (0.49 %) | **28.27 % (0.45 %)** | 7.7H | 37.46 % (0.47 %) | **25.31 % (0.42 %)** | 10.3H | 40.21 % (0.49 %) | **24.75 % (0.44 %)** | 10.3H |

We follow the experimental setting of AQ in (Goldblum et al., 2020), training the state-of-the-art meta-learning models including PROTONET (Snell et al., 2017a), R2D2 (Bertinetto et al., 2018a) , and MetaOptNet ( ResNet-12 as backbone (He et al., 2016)). We conduct all the experiments on the Pytorch with Ubuntu 20.04 and GPU RTX3090.

### 4.2. Experimental results

The experimental results can be seen in Tab. 1. Due to the space limit, training details and other results can be seen in Appendix A.1. To sum up, our LCAT achieves superior performance both on the clean and adversarial few-shot classification accuracy than SOTA adversarial training methods for meta-learning models. For example, we improve 9.7% clean few-shot classification accuracy compared to previous best results in MetaOptNet and improve 2.88% adversarial few-shot classification accuracy compared to previous best results in MetaOptNet corresponding to 5-way 5-shot on MiniImageNet dataset.

## 5. Ablation Study

We also conducted the following experiments: 1) short-term cross adversarial training (SCAT): we set ten epochs as a loop, the first nine epochs are norm training, and the last epoch is adversarial training (T=1). 2) Long-term cross adversarial training (LCAT): we set ten epochs as a loop, the first five epochs are norm training, and the last five epochs are adversarial training (T=5). 3) LCAT with/without denoise (w/o denoise). The results are shown in Tab. 2. which can be seen that our LCAT compare to short-term and (w/o denoise ) achieve supervisor performance.

In addition, we also test the adversarial robustness of meta-learning models trained with LCAT. With the increase of attack step, PROTONET trained with LCAT still has higher adversarial few-shot classification accuracy compared to other adversarial training methods in Fig. 3 (See Appendix A.4 for R2D2 and MetaOptNet ). Further investigations about the influence of TRADES are shown in Tab. 10 in

Appendix A.2.

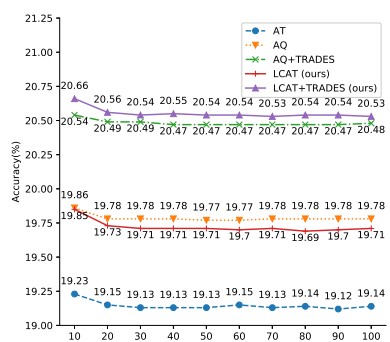

*Figure 3.* $Acc_{adv}$ of five methods on MiniImageNet dataset versus PGD attack at different attack steps on ProtoNet training with 50 epoch

*Table 2.* 5-way, 1-shot MiniImageNet results for ProtoNet, $Acc_{nat}$ and $Acc_{adv}$ stand for natural and robust accuracy of the model, respectively. The top accuracy is bolded.

| METHODS | $Acc_{nat}$ | $Acc_{adv}$ | TIME |
|---|---|---|---|
| SCAT (T=1,EPOCH = 50) | 25.71 % (0.38 %) | 17.97 % (0.32 %) | **3.3H** |
| LCAT (W/O DENOISE) | 31.46 % (0.48 %) | 19.16 % (0.40 %) | 5.7 H |
| LCAT (T=5, EPOCH =50) | **32.55 % (0.49 %)** | **19.72 % (0.41 %)** | 6.6H |
| SCAT (T=1,EPOCH = 100) | 24.29 % (0.36 %) | 18.02 % (0.30 %) | **6.6H** |
| LCAT (W/O DENOISE) | 34.47 % (0.51 %) | 19.37 % (0.41 %) | 11.3H |
| LCAT (T=5, EPOCH =100) | **35.29 % (0.51 %)** | **19.65 % (0.41 %)** | 13.4H |

## 6. Conclusion

In this paper, we propose an adversarially robust meta-learning method called LCAT. LCAT is a model-agonist meta-learning method, which can improve the adversarial robustness of meta-learning models. LCAT will update meta-learning model parameters cross along the natural and adversarial sample distribution direction with long-term to improve both adversarial and clean few-shot classification accuracy. In addition, LCAT only needs half of the adversarial training epoch compared to AQ via cross adversarial training, resulting in a low adversarial training computation. In the future, we will establish a more in-depth understanding of the theoretical analysis of LCAT.

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

# A. Appendix

The code to reproduce all the experimental results is available in `https://github.com/Gnomeek/Long-term-Cross-Adversarial-Training`.

## A.1. Experiments result with models trained in 50/100 epoch on MiniImageNet, TieredImageNet, CIFAR-FS, and FC100

**Training details.** We adopt models including PROTONET (Snell et al., 2017b), R2D2 (Bertinetto et al., 2018b), and MetaOptNet (Lee et al., 2019) and dataset including MiniImageNet (Vinyals et al., 2016), TieredImageNet (Ren et al., 2018), CIFAR-FS (Bertinetto et al., 2018b), and FC100 (Oreshkin et al., 2018). We follow the experimental setting (Goldblum et al., 2020) that the maximum adversarial perturbation magnitude $\epsilon$ is 8/255. In training, the attacker is 7-step PGD with a 2/255 step size. We use Adam (Kingma & Ba, 2014) to optimize the meta-learning models with learning 0.1 and 50 epochs. In testing, we use a 20-step PGD attacker with a 2/255 step size.

*Table 3.* Experiments results of 100 epoch on MiniImageNet dataset with three models, $Acc_{nat}$ and $Acc_{adv}$ stand for natural and robust accuracy of the model, respectively. Robust accuracy is computed for a 20-step PGD attack on the model. Top accuracy scores and the shortest training time are shown in bold font.

| DATASET | METHODS | PROTONET | | | R2D2 | | | METAOPTNET | | |
|---|---|---|---|---|---|---|---|---|---|---|
| | | $Acc_{nat}$ | $Acc_{adv}$ | TIME | $Acc_{nat}$ | $Acc_{adv}$ | TIME | $Acc_{nat}$ | $Acc_{adv}$ | TIME |
| MINIIMAGENET | AT (5WAY-1SHOT) | 34.22 % (0.50 %) | 20.44 % (0.41 %) | 36.3H | 29.03 % (0.43 %) | 19.05 % (0.36 %) | 37.8H | 36.92 % (0.55 %) | 11.59 % (0.35 %) | 46.6H |
| | AQ (5WAY-1SHOT) | 33.61 % (0.48 %) | 20.44 % (0.39 %) | 16.9H | **29.35 % (0.45 %)** | 18.73 % (0.38 %) | 20.6H | 32.09 % (0.46 %) | 11.89 % (0.32 %) | **18.3H** |
| | AQ+TRADES (5WAY-1SHOT) | 30.97 % (0.46 %) | **21.04 % (0.39 %)** | 17.6H | 28.17 % (0.41 %) | **19.36 % (0.37 %)** | 27.7H | 32.73 % (0.48 %) | 15.97 % (0.36 %) | 26.7H |
| | LCAT (OURS, 5WAY-1SHOT) | **35.29 % (0.51 %)** | 19.65 % (0.41 %) | 13.4H | 26.47 % (0.38 %) | 18.67 % (0.33 %) | 15.9H | **39.71 % (0.54 %)** | 12.51% (0.34 %) | 19.2H |
| | LCAT+TRADES (OURS, 5WAY-1SHOT) | 29.58 % (0.43 %) | 18.79 % (0.37 %) | 17.2H | 25.88 % (0.39%) | 18.39 % (0.35 %) | 23.1H | 31.24 % (0.46 %) | **18.51 % (0.39 %)** | 23.6H |
| | AT (5WAY-5SHOT) | 48.76 % (0.54 %) | **29.36 % (0.49 %)** | 36.3H | 41.31 % (0.50 %) | 25.58 % (0.45 %) | 37.8H | 51.07 % (0.51 %) | 17.67 % (0.43 %) | 46.6H |
| | AQ (5WAY-5SHOT) | 47.73 % (0.53 %) | 29.13 % (0.48 %) | 16.9H | 41.31 % (0.50 %) | **42.13 % (0.51 %)** | 20.6H | 45.16 % (0.51 %) | 16.78 % (0.40 %) | **18.3H** |
| | AQ+TRADES (5WAY-5SHOT) | 42.30 % (0.49 %) | 28.61 % (0.44 %) | 17.6H | 37.21 % (0.45 %) | 25.20 % (0.42 %) | 27.7H | 44.11 % (0.50 %) | 22.79 % (0.43 %) | 26.7H |
| | LCAT (OURS, 5WAY-5SHOT) | **48.93 % (0.53 %)** | 29.08 % (0.49 %) | 13.4H | 39.77 % (0.49 %) | 25.06 % (0.43 %) | 15.9H | **54.91 % (0.50 %)** | 18.92 % (0.41 %) | 19.2H |
| | LCAT+TRADES (OURS, 5WAY-5SHOT) | 39.53 % (0.48 %) | 25.38 % (0.43 %) | 17.2H | 32.28 % (0.42 %) | 22.35 % (0.38 %) | 23.1H | 42.49 % (0.51 %) | **26.18 % (0.46 %)** | 23.6H |

*Table 4.* Experiments results of 50 epoch on TieredImageNet dataset with three models, $Acc_{nat}$ and $Acc_{adv}$ stand for natural and robust accuracy of the model, respectively. Robust accuracy is computed for a 20-step PGD attack on the model. Top accuracy scores and the shortest training time are shown in bold font.

| DATASET | METHODS | PROTONET | | | R2D2 | | | METAOPTNET | | |
|---|---|---|---|---|---|---|---|---|---|---|
| | | $Acc_{nat}$ | $Acc_{adv}$ | TIME | $Acc_{nat}$ | $Acc_{adv}$ | TIME | $Acc_{nat}$ | $Acc_{adv}$ | TIME |
| TIEREDIMAGENET | AT (5WAY-1SHOT) | 30.88 % (0.52 %) | **23.93 % (0.51 %)** | 20.9H | 29.31 % (0.50 %) | 22.95 % (0.47 %) | 15.2H | **47.39 % (0.66 %)** | 20.01 % (0.55 %) | 24.7H |
| | AQ (5WAY-1SHOT) | 36.71 % (0.59 %) | 23.19 % (0.50 %) | 8.7H | **35.11 % (0.55 %)** | 23.85 % (0.51 %) | 15.2H | 44.16 % (0.63 %) | 20.73 % (0.54 %) | 10.8H |
| | AQ+TRADES (5WAY-1SHOT) | 34.41 % (0.55 %) | 23.79 % (0.49 %) | 11.0H | 33.49 % (0.55 %) | **24.49 % (0.49 %)** | 17.5H | 40.26 % (0.60 %) | **24.22 % (0.56 %)** | 13.7H |
| | LCAT (OURS, 5WAY-1SHOT) | **36.87 % (0.58 %)** | 22.02 % (0.49 %) | **6.6H** | 33.41 % (0.55 %) | 23.13 % (0.49 %) | **10.0H** | 44.39 % (0.64 %) | 20.48 % (0.53 %) | 10.3H |
| | LCAT+TRADES (OURS, 5WAY-1SHOT) | 34.88 % (0.56 %) | 21.71 % (0.50 %) | 10.0H | 30.71 % (0.50 %) | 21.99 % (0.45 %) | 10.4H | 36.91 % (0.60 %) | 23.38 % (0.54 %) | **10.1H** |
| | AT (5WAY-5SHOT) | 39.83 % (0.53 %) | 30.69 % (0.52 %) | 20.9H | 38.21 % (0.53 %) | 29.66 % (0.52 %) | 15.2H | **62.51 % (0.55 %)** | 29.68 % (0.58 %) | 24.7H |
| | AQ (5WAY-5SHOT) | 52.62 % (0.59 %) | 34.31 % (0.58 %) | 8.7H | **48.52 % (0.56 %)** | 33.61 % (0.56 %) | 15.2H | 58.80 % (0.56 %) | 30.07 % (0.56 %) | 10.8H |
| | AQ+TRADES (5WAY-5SHOT) | 48.01 % (0.58 %) | 34.19 % (0.57 %) | 11.0H | 46.63 % (0.55 %) | **34.06 % (0.56 %)** | 17.5H | 53.54 % (0.54 %) | **34.06 % (0.56 %)** | 13.7H |
| | LCAT (OURS, 5WAY-5SHOT) | **52.90 % (0.59 %)** | 33.39 % (0.58 %) | **6.6H** | 47.29 % (0.55 %) | 32.52 % (0.56 %) | **10.0H** | 59.49 % (0.55 %) | 30.28 % (0.57 %) | 10.3H |
| | LCAT+TRADES (OURS, 5WAY-5SHOT) | 49.07 % (0.58 %) | **34.51 % (0.58 %)** | 10.0H | 42.02 % (0.54 %) | 30.12 % (0.54 %) | 10.4H | 48.47 % (0.54 %) | 32.23 % (0.56 %) | **10.1H** |

*Table 5.* Experiments results of 100 epoch on TieredImageNet dataset with three models, $Acc_{nat}$ and $Acc_{adv}$ stand for natural and robust accuracy of the model, respectively. Robust accuracy is computed for a 20-step PGD attack on the model. Top accuracy scores and the shortest training time are shown in bold font.

| DATASET | METHODS | PROTONET | | | R2D2 | | | METAOPTNET | | |
|---|---|---|---|---|---|---|---|---|---|---|
| | | $Acc_{nat}$ | $Acc_{adv}$ | TIME | $Acc_{nat}$ | $Acc_{adv}$ | TIME | $Acc_{nat}$ | $Acc_{adv}$ | TIME |
| TIEREDIMAGENET | AT (5WAY-1SHOT) | **38.05 % (0.61 %)** | **25.81 % (0.57 %)** | 38.4H | 33.19 % (0.54 %) | 23.33 % (0.50 %) | 32.0H | **47.34 % (0.65 %)** | 19.07 % (0.53 %) | 48.4H |
| | AQ (5WAY-1SHOT) | 37.26 % (0.59 %) | 23.25 % (0.51 %) | 17.7H | **34.37 % (0.54 %)** | 23.73 % (0.50 %) | 30.8H | 44.98 % (0.64 %) | 18.94 % (0.52 %) | 22.9H |
| | AQ+TRADES (5WAY-1SHOT) | 34.90 % (0.55 %) | 23.71 % (0.50 %) | 21.9H | 33.50 % (0.55 %) | **24.51 % (0.50 %)** | 34.0H | 40.65 % (0.60 %) | **24.77 % (0.56 %)** | 26.7H |
| | LCAT (OURS, 5WAY-1SHOT) | 37.20 % (0.58 %) | 21.87 % (0.50 %) | 13.2H | 32.11 % (0.53 %) | 22.82 % (0.46 %) | 18.1H | 45.74 % (0.65 %) | 18.06 % (0.50 %) | **17.5H** |
| | LCAT+TRADES (OURS, 5WAY-1SHOT) | 35.38 % (0.56 %) | 23.73 % (0.50 %) | 20.5H | 32.82 % (0.55 %) | 22.61 % (0.51 %) | 18.9H | 37.31 % (0.59 %) | 23.09 % (0.53 %) | 18.3H |
| | AT (5WAY-5SHOT) | 51.53 % (0.56 %) | **36.05 % (0.57 %)** | 38.4H | 46.13 % (0.56 %) | 32.36 % (0.55 %) | 32.0H | **62.55 % (0.55 %)** | 28.58 % (0.57 %) | 48.4H |
| | AQ (5WAY-5SHOT) | 53.82 % (0.60 %) | 34.76 % (0.59 %) | 17.7H | **47.75 % (0.56 %)** | 33.29 % (0.56 %) | 30.8H | 59.69 % (0.57 %) | 27.97 % (0.54 %) | 22.9H |
| | AQ+TRADES (5WAY-5SHOT) | 48.91 % (0.58 %) | 34.38 % (0.58 %) | 21.9H | 46.55 % (0.56 %) | **34.04 % (0.55 %)** | 34.0H | 53.57 % (0.56 %) | **34.71 % (0.57 %)** | 26.7H |
| | LCAT (OURS, 5WAY-5SHOT) | **54.10 % (0.59 %)** | 33.80 % (0.58 %) | 13.2H | 47.49 % (0.56 %) | 32.73 % (0.57 %) | 18.1H | 61.34 % (0.55 %) | 27.25 % (0.55 %) | **17.5H** |
| | LCAT+TRADES (OURS, 5WAY-5SHOT) | 50.66 % (0.58 %) | 35.22 % (0.58 %) | 20.5H | 42.34 % (0.55 %) | 30.40 % (0.55 %) | 18.9H | 49.70 % (0.56 %) | 32.89 % (0.56 %) | 18.3H |

*Table 6.* Experiments results of 50 epoch on CIFAR-FS dataset with three models, $Acc_{nat}$ and $Acc_{adv}$ stand for natural and robust accuracy of the model, respectively. Robust accuracy is computed for a 20-step PGD attack on the model. Top accuracy scores and the shortest training time are shown in bold font.

| DATASET | METHODS | PROTONET | | | R2D2 | | | METAOPTNET | | |
|---|---|---|---|---|---|---|---|---|---|---|
| | | $Acc_{nat}$ | $Acc_{adv}$ | TIME | $Acc_{nat}$ | $Acc_{adv}$ | TIME | $Acc_{nat}$ | $Acc_{adv}$ | TIME |
| CIFAR-FS | AT (5WAY-1SHOT) | 38.11 % (0.62 %) | **28.42 % (0.59 %)** | 12.2H | 37.11 % (0.62 %) | 27.63 % (0.57 %) | 12.7H | 39.49 % (0.64 %) | 25.61 % (0.61 %) | 17.4H |
| | AQ (5WAY-1SHOT) | **43.22 % (0.66 %)** | 25.94 % (0.62 %) | 2.5H | **41.03 % (0.62 %)** | 28.93 % (0.60 %) | 5.0H | 47.70 % (0.69 %) | 25.44 % (0.65 %) | 6.4H |
| | AQ+TRADES (5WAY-1SHOT) | 39.58 % (0.63 %) | 27.39 % (0.59 %) | 3.1H | 38.89 % (0.61 %) | **29.66 % (0.60 %)** | 6.7H | 42.72 % (0.69 %) | 30.09 % (0.66 %) | 6.4H |
| | LCAT (OURS, 5WAY-1SHOT) | 42.26 % (0.65 %) | 25.67 % (0.61 %) | **2.5H** | 40.43 % (0.64 %) | 28.26 % (0.60 %) | **3.8H** | **48.41 % (0.68 %)** | 25.19 % (0.64 %) | **3.8H** |
| | LCAT+TRADES (OURS, 5WAY-1SHOT) | 38.52 % (0.63 %) | 26.39 % (0.59 %) | 3.8H | 34.68 % (0.58 %) | 26.28 % (0.54 %) | 6.3H | 40.22 % (0.66 %) | **30.42 % (0.64 %)** | 7.5H |
| | AT (5WAY-5SHOT) | 52.02 % (0.59 %) | 38.11 % (0.62 %) | 12.2H | 49.56 % (0.59 %) | 37.51 % (0.59 %) | 12.7H | 52.32 % (0.58 %) | 34.70 % (0.60 %) | 17.4H |
| | AQ (5WAY-5SHOT) | **62.59 % (0.62 %)** | 38.11 % (0.62 %) | 2.5H | **54.86 % (0.59 %)** | **39.87 % (0.59 %)** | 5.0H | 63.82 % (0.58 %) | 36.73 % (0.67 %) | 6.4H |
| | AQ+TRADES (5WAY-5SHOT) | 56.30 % (0.62 %) | **40.76 % (0.64 %)** | 3.1H | 50.95 % (0.58 %) | 39.75 % (0.59 %) | 6.7H | 55.66 % (0.59 %) | 40.39 % (0.63 %) | 6.4H |
| | LCAT (OURS, 5WAY-5SHOT) | 60.29 % (0.61 %) | 38.50 % (0.65 %) | **2.5H** | 53.34 % (0.61 %) | 38.88 % (0.61 %) | **3.8H** | **64.76 % (0.59 %)** | 36.11 % (0.66 %) | **3.8H** |
| | LCAT+TRADES (OURS, 5WAY-5SHOT) | 52.21 % (0.62 %) | 36.50 % (0.62 %) | 3.8H | 46.93 % (0.58 %) | 36.77 % (0.57 %) | 6.3H | 52.87 % (0.64 %) | **41.30 % (0.64 %)** | 7.5H |

*Table 7.* Experiments results of 100 epoch on CIFAR-FS dataset with three models, $Acc_{nat}$ and $Acc_{adv}$ stand for natural and robust accuracy of the model, respectively. Robust accuracy is computed for a 20-step PGD attack on the model. Top accuracy scores and the shortest training time are shown in bold font.

| DATASET | METHODS | PROTONET | | | R2D2 | | | METAOPTNET | | |
|---|---|---|---|---|---|---|---|---|---|---|
| | | $Acc_{nat}$ | $Acc_{adv}$ | TIME | $Acc_{nat}$ | $Acc_{adv}$ | TIME | $Acc_{nat}$ | $Acc_{adv}$ | TIME |
| CIFAR-FS | AT (5WAY-1SHOT) | 42.67 % (0.65 %) | **29.78 % (0.62 %)** | 20.4H | 40.24 % (0.64 %) | 29.11 % (0.61 %) | 22.9H | 42.51 % (0.67 %) | 23.55 % (0.62 %) | 27.8H |
| | AQ (5WAY-1SHOT) | **43.75 % (0.65 %)** | 25.77 % (0.61 %) | 5.4H | 41.44 % (0.63 %) | 28.78 % (0.59 %) | 10.6H | 48.76 % (0.67 %) | 23.18 % (0.61 %) | 14.1H |
| | AQ+TRADES (5WAY-1SHOT) | 39.87 % (0.64 %) | 27.27 % (0.59 %) | 6.4H | 39.41 % (0.62 %) | **30.26 % (0.60 %)** | 12.4H | 42.24 % (0.65 %) | **29.73 % (0.64 %)** | 13.9H |
| | LCAT (OURS, 5WAY-1SHOT) | 42.51 % (0.65 %) | 24.95 % (0.61 %) | **4.9H** | **41.49 % (0.62 %)** | 28.27 % (0.59 %) | **7.3H** | **50.29 % (0.70 %)** | 24.13 % (0.63 %) | **9.4H** |
| | LCAT+TRADES (OURS, 5WAY-1SHOT) | 37.94 % (0.63 %) | 25.75 % (0.58 %) | 7.5H | 35.88 % (0.61 %) | 27.85 % (0.56 %) | 13.9H | 41.15 % (0.65 %) | 29.51 % (0.62 %) | 13.9H |
| | AT (5WAY-5SHOT) | 58.59 % (0.59 %) | **41.37 % (0.63 %)** | 20.4H | 53.06 % (0.60 %) | 39.33 % (0.60 %) | 22.9H | 56.81 % (0.59 %) | 33.39 % (0.62 %) | 27.8H |
| | AQ (5WAY-5SHOT) | **63.66 % (0.60 %)** | 39.78 % (0.67 %) | 5.4H | **55.59 % (0.59 %)** | 39.83 % (0.60 %) | 10.6H | 65.84 % (0.57 %) | 34.23 % (0.63 %) | 14.1H |
| | AQ+TRADES (5WAY-5SHOT) | 57.44 % (0.62 %) | 41.25 % (0.65 %) | 6.4H | 51.52 % (0.58 %) | **40.27 % (0.59 %)** | 12.4H | 55.45 % (0.60 %) | 40.12 % (0.62 %) | 13.9H |
| | LCAT (OURS, 5WAY-5SHOT) | 61.69 % (0.60 %) | 38.51 % (0.66 %) | **4.9H** | 54.97 % (0.60 %) | 39.28 % (0.61 %) | **7.3H** | **66.95 % (0.58 %)** | 35.22 % (0.66 %) | **9.4H** |
| | LCAT+TRADES (OURS, 5WAY-5SHOT) | 51.14 % (0.62 %) | 36.50 % (0.62 %) | 7.5H | 47.14 % (0.59 %) | 37.37 % (0.59 %) | 13.9H | 55.72 % (0.61 %) | **41.33 % (0.64 %)** | 13.9H |

*Table 8.* Experiments results of 50 epoch on FC-100 dataset with three models, $Acc_{nat}$ and $Acc_{adv}$ stand for natural and robust accuracy of the model, respectively. Robust accuracy is computed for a 20-step PGD attack on the model. Top accuracy scores and the shortest training time are shown in bold font.

| DATASETS | METHODS | PROTONET | | | R2D2 | | | METAOPTNET | | |
|---|---|---|---|---|---|---|---|---|---|---|
| | | $Acc_{nat}$ | $Acc_{adv}$ | TIME | $Acc_{nat}$ | $Acc_{adv}$ | TIME | $Acc_{nat}$ | $Acc_{adv}$ | TIME |
| FC100 | AT(5WAY-1SHOT) | 32.32 % (0.51 %) | 20.71 % (0.46 %) | 5.2H | **33.26 % (0.55 %)** | 21.31 % (0.51 %) | 6.3H | 32.04 % (0.50 %) | 16.08 % (0.46 %) | 10.7H |
| | AQ(5WAY-1SHOT) | 32.43 % (0.52 %) | 25.75 % (0.47 %) | 4.8H | 33.21 % (0.54 %) | 21.54 % (0.50 %) | 5.3H | 32.47 % (0.51 %) | 15.87 % (0.46 %) | 5.7H |
| | AQ+TRADES(5WAY-1SHOT) | 30.54 % (0.47 %) | 21.83 % (0.43 %) | 3.0H | 31.69 % (0.53 %) | **22.65 % (0.50 %)** | 2.4H | **33.28 % (0.57 %)** | **22.29 % (0.53 %)** | 5.3H |
| | LCAT(OURS, 5WAY-1SHOT) | **34.77 % (0.54 %)** | 19.69 % (0.49 %) | 3.4H | 32.60 % (0.52 %) | 20.53 % (0.48 %) | 4.0H | 33.02 % (0.50 %) | 13.40 % (0.41 %) | **4.8H** |
| | LCAT+TRADES(OURS, 5WAY-1SHOT) | 31.86 % (0.51 %) | **22.00 % (0.48 %)** | **3.0H** | 31.18 % (0.51 %) | 22.42 % (0.48 %) | **2.4H** | 30.94% (0.50%) | 21.49% (0.46%) | 6.5H |
| | AT(5WAY-5SHOT) | 43.41 % (0.54 %) | 27.59 % (0.54 %) | 5.2H | 42.87 % (0.55 %) | 28.03 % (0.55 %) | 6.3H | 41.82 % (0.52 %) | 21.56 % (0.51 %) | 10.7H |
| | AQ(5WAY-5SHOT) | 43.30 % (0.55 %) | 27.28 % (0.53 %) | 4.8H | **42.93 % (0.54 %)** | 28.29 % (0.54 %) | 5.3H | 42.58 % (0.52 %) | 21.09 % (0.50 %) | 5.7H |
| | AQ+TRADES(5WAY-5SHOT) | 40.96 % (0.53 %) | 29.01 % (0.52 %) | 3.0H | 40.14 % (0.53 %) | **29.06 % (0.53 %)** | 2.4H | 40.70 % (0.54 %) | **27.90 % (0.53 %)** | 5.3H |
| | LCAT(OURS, 5WAY-5SHOT) | **45.17 % (0.55 %)** | 26.95 % (0.54 %) | 3.4H | 41.89 % (0.54 %) | 27.42 % (0.53 %) | 4.0H | **43.54 % (0.51 %)** | 19.10 % (0.47 %) | **4.8H** |
| | LCAT+TRADES(OURS, 5WAY-5SHOT) | 41.28 % (0.55 %) | **29.31 % (0.54 %)** | **3.0H** | 39.17 % (0.54 %) | 28.25 % (0.53 %) | **2.4H** | 39.61 % (0.55%) | 27.72 % (0.53%) | 6.5H |

*Table 9.* Experiments results of 100 epoch on FC-100 dataset with three models, $Acc_{nat}$ and $Acc_{adv}$ stand for natural and robust accuracy of the model, respectively. Robust accuracy is computed for a 20-step PGD attack on the model. Top accuracy scores and the shortest training time are shown in bold font.

| DATASETS | METHODS | PROTONET | | | R2D2 | | | METAOPTNET | | |
|---|---|---|---|---|---|---|---|---|---|---|
| | | $Acc_{nat}$ | $Acc_{adv}$ | TIME | $Acc_{nat}$ | $Acc_{adv}$ | TIME | $Acc_{nat}$ | $Acc_{adv}$ | TIME |
| FC100 | AT(5WAY-1SHOT) | 33.60 % (0.52 %) | 19.15 % (0.46 %) | 10.5H | 33.46 % (0.55 %) | 20.73 % (0.50 %) | 13.9H | 32.86 % (0.49 %) | 12.94 % (0.39 %) | 21.2H |
| | AQ(5WAY-1SHOT) | 33.67 % (0.52 %) | 19.83 % (0.47 %) | 8.1H | 33.30 % (0.53 %) | 20.39 % (0.49 %) | 10.0H | 32.63 % (0.49 %) | 14.20 % (0.43 %) | 13.0H |
| | AQ+TRADES(5WAY-1SHOT) | 31.72 % (0.49 %) | **22.03 % (0.45 %)** | 5.5H | 32.93 % (0.54 %) | **22.51 % (0.52 %)** | 4.8H | 32.88 % (0.57 %) | **21.10 % (0.52 %)** | 11.6H |
| | LCAT(OURS, 5WAY-1SHOT) | **35.28 % (0.55 %)** | 18.42 % (0.49 %) | 7.6H | **34.11 % (0.54 %)** | 20.35 % (0.50 %) | 7.5H | **34.09 % (0.50 %)** | 10.72 % (0.39 %) | **10.6H** |
| | LCAT+TRADES(OURS, 5WAY-1SHOT) | 31.44 % (0.50 %) | 21.34 % (0.46 %) | **5.4H** | 31.22 % (0.51 %) | 21.75 % (0.47 %) | **4.8H** | 31.94% (0.53%) | 20.80% (0.48%) | 13.1H |
| | AT(5WAY-5SHOT) | 46.05 % (0.55 %) | 26.42 % (0.54 %) | 10.5H | 43.40 % (0.56 %) | 27.27 % (0.55 %) | 13.9H | 44.15 % (0.54 %) | 18.67 % (0.49 %) | 21.2H |
| | AQ(5WAY-5SHOT) | 46.17 % (0.55 %) | 27.16 % (0.53 %) | 8.1H | **43.75 % (0.55 %)** | 27.33 % (0.55 %) | 10.0H | 43.52 % (0.53 %) | 19.54 % (0.51 %) | 13.0H |
| | AQ+TRADES(5WAY-5SHOT) | 42.18 % (0.54 %) | **28.93 % (0.53 %)** | 5.5H | 41.65 % (0.53 %) | **29.06 % (0.54 %)** | 4.8H | 40.65 % (0.54 %) | 26.54 % (0.53 %) | 11.6H |
| | LCAT(OURS, 5WAY-5SHOT) | **46.66 % (0.57 %)** | 25.91 % (0.55 %) | 7.6H | 43.68 % (0.55 %) | 27.12 % (0.54 %) | 7.5H | **44.92 % (0.54 %)** | 15.35 % (0.46 %) | **10.6H** |
| | LCAT+TRADES(OURS, 5WAY-5SHOT) | 41.17 % (0.53 %) | 28.72 % (0.53 %) | **5.4H** | 39.69 % (0.53 %) | 28.35 % (0.53 %) | **4.8H** | 40.98 % (0.55%) | **27.13 % (0.54%)** | 13.1H |

## A.2. Experiments results on applying TRADES at different times

*Table 10.* Experiments results on MiniImageNet dataset with ProtoNet model, $Acc_{nat}$ and $Acc_{adv}$ stand for natural and robust accuracy of the model, respectively. TRADE$_{10}^{10}$ and TRADE$_{10}^{5}$ represent using TRADES during all the epoch, last 5 epoch in every 10 epoch, respectively, i.e., using TRADES all the time or using it only when attack method is applied. Robust accuracy is computed for a 20-step PGD attack on the model. Top accuracy scores and the shortest training time are shown in bold font.

| METHODS | $Acc_{nat}$ | $Acc_{adv}$ | TIME |
|---|---|---|---|
| LCAT+TRADES$_{10}^{10}$ (EPOCH =50) | 30.23 % (0.45 %) | 19.81 % (0.39 %) | 8.9H |
| LCAT+TRADES$_{10}^{5}$ (EPOCH = 50) | **30.71 % (0.46 %)** | **20.56 % (0.39 %)** | **7.7H** |
| LCAT+TRADES$_{10}^{10}$ (EPOCH = 100) | 29.19 % (0.43 %) | **19.00 % (0.37 %)** | 17.5H |
| LCAT+TRADES$_{10}^{5}$ (EPOCH = 100) | **29.58 % (0.43 %)** | 18.79 % (0.37 %) | **17.2H** |

## A.3. Experiments results on LCAT applied with different optimizer

*Table 11.* Experiments results on MiniImageNet dataset with ProtoNet model, $Acc_{nat}$ and $Acc_{adv}$ stand for natural and robust accuracy of the model, respectively. Robust accuracy is computed for a 20-step PGD attack on the model. Top accuracy scores and the shortest training time are shown in bold font.

| METHODS | $Acc_{nat}$ | $Acc_{adv}$ | TIME |
|---|---|---|---|
| LCAT (ADAM, EPOCH =50) | **32.55 % (0.49 %)** | **19.72 % (0.41 %)** | **6.6H** |
| LCAT (SGD, EPOCH = 50) | 32.48 % (0.48 %) | 18.50 % (0.39 %) | 8.2H |
| LCAT (ADAM, EPOCH = 100) | **35.29 % (0.51 %)** | **19.65 % (0.41 %)** | **13.4H** |
| LCAT (SGD, EPOCH = 100) | 31.41 % (0.46 %) | 17.88 % (0.37 %) | 16.1H |

## A.4. Extended experiments results on R2D2 and MetaOptNet versus PGD attack at different attack steps

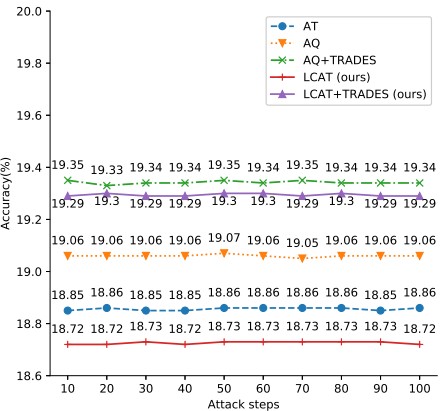

*Figure 4.* $Acc_{adv}$ of five methods on MiniImageNet dataset versus PGD attack at different attack steps on R2D2 training with 50 epoch

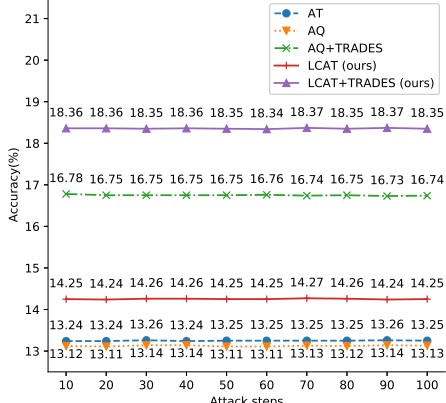

*Figure 5.* $Acc_{adv}$ of five methods on MiniImageNet dataset versus PGD attack at different attack steps on MetaOptNet training with 50 epoch