# OpenReview forum: "Long-term Cross Adversarial Training: A Robust Meta-learning Method for Few-shot Classification Tasks "
_ICML.cc/2021/Workshop/AML — ICML 2021 Workshop AML Poster_

### Official Review · Reviewer_Vez6 · 2021-06-19
**The proposed method is promising. More theoretical analysis should be given. Accept.**

**Rating:** Accept
**Confidence:** 4

**Review:**

This paper proposed a meta-learning method on the adversarially robust neural network. The method is called Long-term Cross Adversarial Training (LCAT). The proposed LCAT will update metalearning model parameters cross along the natural and adversarial sample distribution direction with long-term to improve both adversarial and clean few-shot classification accuracy.

Pros: The proposed method is computationally expensive. The performance is better than SOTA. Experiment results are promising.

Cons: More theoretical analysis should be given about the proposed method.

---

### Decision · Program_Chairs · 2021-06-21

**Decision:**

Accept (Poster)

**Comment:**

This paper proposed a robust meta-learning method for few-shot classification. The performance is better and the results are promising.